# Valuable Fatty Acids in Bryophytes—Production, Biosynthesis, Analysis and Applications

**DOI:** 10.3390/plants8110524

**Published:** 2019-11-19

**Authors:** Yi Lu, Finnur Freyr Eiriksson, Margrét Thorsteinsdóttir, Henrik Toft Simonsen

**Affiliations:** 1Department of Biotechnology and Biomedicine, Technical University of Denmark, Søltofts Plads 223, 2800 Kongens Lyngby, Denmark; yilu@dtu.dk; 2ArcticMass, Sturlugata 8, 101 Reykjavik, Iceland; finnur@arcticmass.is (F.F.E.); margreth@hi.is (M.T.); 3Faculty of Pharmaceutical Sciences, University of Iceland, Hagi, Hofsvallagata 53, 107 Reykjavik, Iceland

**Keywords:** arachidonic acid, bryophytes, eicosapentaenoic acid, environmental stress, polyunsaturated fatty acid

## Abstract

Bryophytes (mosses, liverworts and hornworts) often produce high amounts of very long-chain polyunsaturated fatty acids (vl-PUFAs) including arachidonic acid (AA, 20:4 Δ5,8,11,14) and eicosapentaenoic acid (EPA, 20:5 Δ5,8,11,14,17). The presence of vl-PUFAs is common for marine organisms such as algae, but rarely found in higher plants. This could indicate that bryophytes did not lose their marine origin completely when they landed into the non-aqueous environment. Vl-PUFA, especially the omega-3 fatty acid EPA, is essential in human diet for its benefits on healthy brain development and inflammation modulation. Recent studies are committed to finding new sources of vl-PUFAs instead of fish and algae oil. In this review, we summarize the fatty acid compositions and contents in the previous studies, as well as the approaches for qualification and quantification. We also conclude different approaches to enhance AA and EPA productions including biotic and abiotic stresses.

## 1. Introduction

Bryophytes are an informal group of three divisions of non-vascular land plant, which consist of about 25,000 known species globally. They are divided into mosses (Bryophyta, 18,000 species), liverworts (Marchantiophyta, 6000 species) and hornworts (Anthocerotophyta, 1000 species) [1]. Bryophytes are amphibian plants, which are found in almost all kinds of habitats world-wide, from dry desert to humid rainforest, from hot tropical area to the cold Arctic and from sea level to alpine peaks [2]. Bryophytes are taxonomically important as they are believed to represent a close connection between aquatic lives to land organisms [3]. Liverworts are probably the earliest land plants, which are believed to be the first landed plant, almost 500 million years ago [4,5].

In general, bryophytes show high tolerance against various biotic and abiotic stresses [6,7]. Unlike higher plants, bryophytes do not have a vascular system and lack thick waxy cutin protection on the cell walls [8], they defend themselves by producing several specialized metabolites instead. This can explain why bryophytes are rarely consumed by insects and herbivores, not only because bryophytes have low caloric content but also for the diversity of “chemical weapons” they produce [9,10]. Besides, bryophytes have a high desiccation tolerance and freezing tolerance, which allow them to survive in dry surroundings or in cold temperatures and recover rapidly afterwards [11].

The chemical composition of bryophytes, from macro-compounds (carbohydrates, proteins and lipids) [12] to specific substance groups such as fatty acids (FAs) [13,14,15], terpenoids [1,16,17], flavonoids [1] and polyphenols [18], have already been studied.

However, when compared to algae and higher plants, the phytochemistry of bryophytes is still poorly understood, possibly because they are morphologically small, and it is also difficult to collect pure samples in large quantities. Alongside the small quantities of material, elimination of all accompanying materials (e.g., dead tree leaves, soil, twigs, even small animals) can be difficult [19]. A possible method to obtain large bryophyte biomass is to grow them in liquid culture in bioreactors. Several in vitro cultivations have been applied on over 300 mosses and 50 liverworts [20], though one has to put a lot of effort into this, in order to obtain purely axenic cultures [21].

Among all metabolites in bryophytes, lipids are considered to be one of the most important groups, as they play many vital roles in energy storage, membrane formation, cell signaling, functioning and environmental adaption [22]. The content of lipids ranges from 1 to 9.1% of dry weight depending on the region and growth condition [23]. In general, lipids refer to all non-hydrophilic compounds, including free fatty acids, triglycerides, glycolipids, phospholipids, sterols, wax esters, fatty alcohols and terpenoids. These lipophilic chemical constituents could be the reason that bryophytes have been used in many countries as ethno-medicine for the treatment of cuts, burns and bruises because of their anti-fungal, anti-inflammatory and anti-oxidant functions [1,10,24,25,26]. The genera *Sphagnum*, *Marchantia* and *Polytrichum* are the most widely used bryophytes worldwide [24], mostly in China, followed by USA and Canada [24], whereas *Physcomitrella patens* is the only one used for industrial biotechnological purposes [11,27,28]. Recent in vivo experiments [29] also tested the antifeedant and anti-fungal activities of extracts from several bryophytes species, indicating a potential value of a promising biopesticide from bryophytes for the replacement of synthesized pesticides. This has already been sold as a commercial product in Germany [30]. The important biologically-active compounds from bryophytes are mainly terpenoids and phenylpropanoid- derived compounds, such as polyphenol and flavonoids. As for the terpenoids, most studies have worked with liverworts because of the presence of oil bodies, where the terpenoids are stored [6]. More than 1600 terpenoids have been identified from liverworts [1]. Currently there are already several reviews on terpenoids [1,16,31,32,33] and phenylpropanoid-derived-compounds [10,31], where the chemical structures of these compounds and their biological activities are explained.

Very long chain polyunsaturated fatty acids (vl-PUFAs) have received a lot of interest in recent years due to their health-promoting effect in humans and livestock [34,35]. Bryophytes produce high contents of arachidonic acid (AA, 20:4, Δ5,8,11,14) and eicosapentaenoic acid (EPA, 20:5, Δ5,8,11,14,17), which are uncommon in higher plant. This suggests that bryophytes are genetically closer to algae rather than other terrestrial plants [36]. Their high ω-3/ω-6 ratio also suggest an alternative and sustainable approach to improve human vl-PUFAs intake by inserting high-PUFAs-bryophyte-genes in order to modify current oilseed genes [37]. Most of the studies in this field have been focused on the model moss species *P. patens* [38] and common liverwort *Marchantia polymorpha* [39], which are the two species with complete genome sequences that have been published in bryophytes.

In this review, we only focus on fatty acids and their linked lipid classes in bryophytes. The purpose of this review is to summarize the production of fatty acids and the accumulation of AA and EPA under various stress conditions in bryophytes from recent published studies. Furthermore, this review also presents strategies for the identification and quantification of fatty acids in bryophytes. Finally, we wish to highlight the potential use of bryophytes as new sources for vl-PUFAs.

## 2. Fatty Acid Biosynthetic Pathways

A brief biosynthetic pathway of major fatty acids produced by bryophytes is shown in Figure 1. Acetyl-CoA and malonyl-CoA are the basic building blocks for fatty acid biosynthesis, followed by a series of reactions by fatty acyl synthases. Acyl carrier proteins (ACPs) carry the intermediates during the fatty acid elongation. The emerging chain is elongated through six cycles to reach the primary product palmitic acid (16:0), and a small amount of shorter chain fatty acids may be released before reaching 16 carbons in length or breakdown products. After releasing from plastid, linoleic acid (LA,18:2, Δ9,12, ω-6) serves as a precursor of a family of PUFAs, which is formed by desaturation and chain elongation [22]. Plants, including bryophytes, can synthesize α-linolenic acid (ALA, Δ9,12,15, ω-3) from LA catalyzed by Δ15 desaturase. Alternatively, LA can also be converted by Δ-6 desaturase to γ-linolenic acid (GLA, 18:3, Δ6,9,12), Δ-6 elongase to di-homo γ-linolenic acid (DGLA, 20:3, Δ8,11,14) and Δ-5 desaturase to AA (ω-6). The enzymes, Δ5- and Δ-6 desaturases are responsible for LA and ALA unsaturation, while the enzyme Δ-6 elongase is responsible for the elongation of C18 fatty acids to C20 fatty acids. Δ-6 desaturase, Δ-6 elongase and Δ-5 desaturase are the keys for high PUFAs production in bryophytes, since they are not expressed in higher plants [40].

In addition to common FAs, odd-chain FAs were found in trace amounts in bryophytes, which may be synthesized when fatty acid synthase accepts propionyl-CoA instead of acetyl-CoA as a primer molecule or when alpha-oxidation of FAs occurs [41].

## 3. Analysis of Fatty Acids in Bryophytes

Different lipid analysis methods have been developed and evaluated on several bryophyte species in the past few decades. To investigate the composition of fatty acids and their linked lipid classes such as neutral lipids (NLs), polar lipids (PLs) or glycolipids (GLs), a step of lipid separation (or purification) is usually performed using thin layer chromatography (TLC) to yield different lipid fractions. Solid phase extraction (SPE) is also a well-developed method for lipid separation, solvents with different polarities are used for separating different lipid classes [42]. Klavina and Kviesis (2015) used the SPE column to separate different groups of lipids in moss *Polytrichum commune* and *Dicranum polysetum* by using hexane for the extraction of alkanes, sterols, the fatty alcohols; hexane/chloroform (5:1), for the extraction of esters, ketones, aromatic substances; and chloroform for extraction of sterols [43].

Since most of the studies in bryophytes have been focusing on volatile specialized metabolites, most of the chemical analyses is performed on gas chromatography-flame ionization detector (GC-FID) and gas chromatography-mass spectrometry (GC-MS) (Table 1). These techniques require relatively large quantities of samples that have to be cleaned from other interfering materials. The cleaning process for bryophytes is time-consuming compared with other plants, as they grow in between each other and other organisms. Thus, obtaining grams of “pure” species is long and tedious work. The need of derivatization and limited resolution along with difficulties in the identification of fatty acids and their isomers also lead to adjacent masses and peak interference [44]. Thus, the development of a new method for bryophyte lipid analysis, a method which only requires milligrams of fresh material, is of great importance. The emerging field in lipidomics is a new platform for lipid-related metabolic pathways in plants. Screening of biomarkers by untargeted lipid analysis using liquid chromatography-mass spectrometry (LC-MS), allows for the separation and identification of possible lipophilic compounds and derivatives in biological samples. It is a powerful tool to evaluate the phylogenetic diversity among different species [45] and examine the change of related lipid species due to environmental variability [46]. The lipid extract can also be injected directly to electrospray ionization triple quadrupole mass spectrometry (ESI-MS/MS), which has a great advantage on analyzing the formation of PLs in plants [46]. Matrix-assisted laser desorption/ionization time-of-flight mass spectrometry (MALDI-TOF-MS) is another analytical technique that has good sensitivity and reproducibility for analyzing plant endogenous molecules [47] and for a rapid screening of free FAs in biological samples from different origins [48]. Generally, LC-based lipidomics is relatively rapid and usually requires less sample than GC-derived methods, since only 2–10 mg of leaf dry weight is needed. So far, lipid profiling has been performed on a large variety of plants such as *Arabidopsis thaliana* [49], and several algae species such as the snow alga *Chlamydomonas nivalis* [50], the diatom *Nitzschia Closterium* [51] and the brown alga *Sargassum horneri* [52], but these advanced techniques have not yet been applied on bryophytes.

## 4. Fatty Acids Present in Bryophytes

FAs are usually present as part of membrane phospho- and glycolipids, or as constituents in triacylglycerides (TAGs); healthy living bryophytes tissue do not normally accumulate free FAs [71]. FAs from bryophytes, including saturated, mono-, poly-unsaturated and acetylenic fatty acids (AFAs) are listed in Table 2. Similar to the other plants and organisms, bryophytes can synthesize some common saturated fatty acids such as palmitic acid (16:0) and stearic acid (18:0). Medium-chain fatty acids, such as lauric acid (12:0) and myristic acid (14:0) are also found in a limited amount. Some odd-chain saturated fatty acids, pentadecanoic acid (15:0) and margaric acid (17:0), which do not commonly appear in nature, can also be found in trace amounts in some bryophytes. Several mono- and polyunsaturated fatty acids, such as oleic acid (OA, 18:1, Δ9), LA and ALA are abundant in all bryophytes species. Long-chain- and very-long-chain fatty acids (C20 and above) are rarely produced by higher plant, but high amount of vl-PUFAs, AA and EPA in particularly, are commonly found in bryophytes. High contents of PUFAs are important for bryophytes to survive under low temperature and harsh environment [72], where some bryophytes can even survive at −14 °C [73]. These differences are of great importance to distinguish bryophytes from other higher plants.

The vl-PUFAs, especially ω-3 PUFAs, are essential for human health as they play an important role in eicosanoids synthesis (e.g., prostaglandins, thromboxanes, leukotrienes and lipoxins), cell signaling and gene expression [34]. Humans de novo synthesize LA and ALA with very low synthetic efficiency, therefore we must obtain these essential fatty acids from food [74]. Although bryophytes are not an ideal source for food consumption due to the poor nutrient content [9], this unique metabolism of bryophytes can still indirectly benefit human by optimizing ω-3 oil production in seed crops by a transgenic approach [37].

AFAs are extensively found in some bryophyte families. Many studies show that AFAs appear as part of triacylglycerol to maximize energy conservation when growth space is limited [23]. The uncommon AFA Dicranin (octadeca-6-yn-9, 12, 15-trienoic acid, 18:4a) is almost found exclusively in the Dicranaceae family. Several studies showed high AFA contents in Dicranaceae species [55,70,75]. In particularly, 72.1% of AFA of total fatty acid was found in *Dicranum polysetum* [76], of which Dicranin was the predominant AFA with 23.5% of total fatty acids. 

AFAs appear to have pharmaceutical properties, and the moss *Rhodobryum* was used as traditional Chinese medicine to treat cardiovascular diseases, possibly due to the richness of ALA and Dicranin [68]. Likewise, ALA is also a precursor of some acetylenic oxylipins, which act as defense compounds against bacteria, fungi and insects [77].

## 5. Arachidonic Acid (AA) and Eicosapentaenoic Acid (EPA)

AA and EPA are rarely found in higher plants [82], however they are both common in bryophtyes. AA and EPA are precursors for the biosynthesis of some eight-carbon fragrant fatty alcohol, such as 1-octen-3-ol, octan-3-one and octan-3-ol, in damaged *M. polymorpha* [83]. Mosses and liverworts with high contents of AA and EPA in proportion to total lipid are shown in Table 3.

VL-PUFAs are essential for human, since they are the main constituents of human brain phospholipids, and can prevent cardiovascular diseases [34]. Human cannot synthesize ALA (ω-3) from its precursor LA (ω-6), since we do not have Δ-15 desaturase, thus, ω-3 and ω-6 fatty acids imbalance is an unignorably problem of human health [35]. Extra ω-3 PUFAs such as EPA and DHA are often taken as daily supplements derived from fish oils. However, the accumulation of heavy metals and reduced production of marine fish make it an unsustainable source of ω-3 vl-PUFA [74]. Consequently, the development of other sources for ω-3 vl-PUFA is in urgent need. A novel, affordable and renewable approach is to transfer the ω-3 vl-PUFA-expressed-genes to oilseed crops by using transgenic engineering [37]. High accumulation of AA (1.6%) and EPA (2.7%) were observed in linseed oil when Δ5, Δ6 desaturases from the diatom *Phaeodactylum tricornutum* and Δ6 elongases from *P. patens* were successfully expressed in linseed [87]. The same three genes from liverwort *M. polymorpha* were introduced into tobacco, which resulted in a production of 15.5% AA and 4.9% EPA of total fatty acids in the leaves [40]. This shows that there is a huge potential for a new source of vl-PUFAs for human consumption in addition to fish oil, although no commercial product is available in the market now.

## 6. Strategies of Enhancing the Production of vl-PUFAs

Only a small proportion of bryophytes species have been examined for their fatty acid composition, the changes of fatty acids composition under biotic and abiotic stress conditions are mostly unexplored. Scientists are still looking for new species with naturally high content of AA and EPA. Efforts have been made to enhance the production of AA and EPA in bryophytes in the past few decades, either by exposing them to different environmental stressors or by transgenic approaches.

### 6.1. Developmental Stages

Different developmental stages of bryophytes have different fatty acid composition. Protonema was richer in AA but less in EPA compare to gametophytes in five moss species (Table 3), but a reverse result was concluded in *P. patens* [13]. However, further studies are needed to understand the changes better.

### 6.2. Environmental Stressors

Environmental factors are keys to enhance PUFA production. PUFAs content and the ω-3/ω-6 ratio are affected by many factors such as temperature, light, pH and nutrition [65,66]. Fatty acids compositions of several mosses and liverworts have been examined under different environmental conditions. When cultivated in cell cultures, the size of inoculum and ferrous ion also affect PUFA productivity [64].

Light is one of the most important factors that affects the accumulation of PUFAs, not only the light intensity but also the light quality [65]. Blue light enhances the accumulation of EPA content in *M. polymorpha* but has little effect on AA content. In *M. polymorpha* cell cultures the optimum light intensity has been found to be 80 photons μmol/m^2^/s [65].

After cold treatments in the cell cultures of the moss *Rhytidiadelphus squarrosus*, *Eurhynchium striatum* [59] and the liverwort *M. polymorpha* [88], all of them showed increase of ω-3 PUFAs (EPA) and decrease of ω-6 PUFAs (AA) although growth deficits were also observed. The low temperature resulted in an increase of EPA; more specifically, EPA content at 5 °C was three times higher than at 25 °C in *M. polymorpha*, indicating ω-3 desaturase gene expression was induced at cold temperatures [88]. During cold stress of *M. polymorpha* a change in EPA was found for monogalactosyldiacylglycerol (MGDG) and chloroplastic phosphatidycholine (PC) [63]. Overall temperature regulation can assist in the production of high levels of PUFAs and can be regulated so that the overall growth rate is not affected.

Nutrients such as nitrogen, sulfur, phosphor and minerals are essential for a healthy bryophyte culture. Both total lipid contents and PUFAs proportion of four moss species *Ctenidium molluscum*, *Pogonatum urnigerum*, *Dichodontium pellucidum* and *Tortella tortuosa* showed a decrease in levels when the NO_3_^-^ concentration was increased in the media (Table 4), indicating that mosses synthesize more proteins rather than lipids when the medium is rich in nitrogen [89]. Thus, in order to obtain moss biomass with high PUFA content, it was suggested to use biphasic process as growing moss on nitrogen-rich medium for high biomass concentration, followed by a nitrogen starvation phase for improving the lipid content. With modern bioreactor technologies this could be linked with temperature control, which will allow the bryophyte to grow fast, and when the nitrogen is used then the temperature can be lowered to induce PUFA production.

### 6.3. Genetic Transformation

All studies of genetic transformation to enhance vl-PUFA production in bryophytes have been focused on *M. polymorpha* [40] and *P. patens* [90,91]. Overexpression of genes encoding enzymes Δ-5 desaturase, Δ-6 desaturase and Δ-6 elongase in *M. polymorpha* resulted in 3- and 2-fold increase of AA and EPA, respectively, compared to those in the wild type [40]. When both Δ-6 elongase from *P. patens* and lipid-linked Δ-5 desaturase from diatom *Phaeodactylum tricornutum* were expressed in marine alga *Saccharomyces cerevisiae*, high proportions of AA and EPA were detected because almost all Δ-6 desaturated products were elongated [92].

## 7. Future Perspectives

Bryophytes are underexploited for their valuable biologically-active compounds. High contents of arachidonic acid and eicosapentenoic acid in bryophytes highlight their potential usage in the pharmaceutical industry, food industry and cosmetics. Efforts have been put on in vitro cultivation of bryophytes in liquid culture in bioreactors in order to obtain both sufficient biomass and high contents of valuable fatty acids. Current studies in the large scale tend to focus on model species the *P. patens*. The techniques of the in vitro cultivation of *P. Patens* can be transferred to other attractive species to meet industrial demands. Lipidomics is a powerful tool to examine lipid-related molecular mechanisms and lipid biomarkers in bryophytes in response to stress conditions.

## Figures and Tables

**Figure 1 plants-08-00524-f001:**
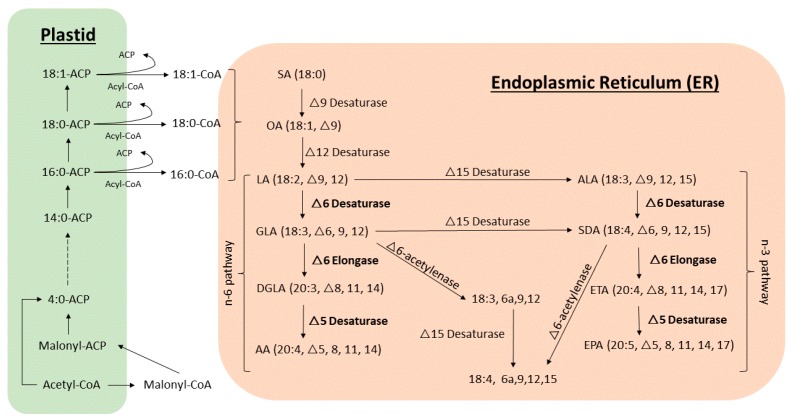
An overview of PUFAs and acetylenic fatty acids synthetic pathways in bryophytes. Fatty acids biosynthesis starts from a consecutive attachment of two carbon units until 16:0, 18:0 and 18:1 in plastid, where the fatty acids are attached to the Acyl carrier protein (ACP), and some enzymes that terminate the chain-elongation early, resulting in the production of fatty acids with shorter chains. 16:0, 18:0 and 18:1 are further transferred to the endoplasmic reticulum (ER) for PUFAs production by several fatty acid desaturases. Some unusual occurrence of acetylenic acids are also synthesized in ER. SA, stearic acid; OA, oleic acid; LA, linoleic acid; GLA, γ-linolenic acid; DGLA, di-homo γ-linolenic acid; AA, arachidonic acid; ALA, α-linolenic acid; SDA, stearidonic acid; ETA, eicosatetraenoic acid; EPA, eicosapentaenoic acid.

**Table 1 plants-08-00524-t001:** Methods used for characterization and quantification of lipids in bryophytes.

Species	Lipid Types	Methods	Derivatization	References
*Mnium cuspidotum*	FAs	TLC for purification, GC for quantification	Alkylation (diazomethane)	[53]
*Ceratodon purpureus*	FAs	GC for purification, Mass spectrometry for molecular weight, UV and IR for structural confirmation	Alkylation (diazomethane)	[54]
38 moss species	FAs	TLC for purification and identification. GC-MS for FAs analysis	Alkylation (methylation)	[55]
*Riccia fluitans*	FAs	GC for purification, FAs analysis by GC-MS, NMR	Alkylation (methylation)	[56]
Twelve liverworts in genus *Riccia*	FAs	TLC for purification and identification, GC-MS for analysis	Alkylation (methylation)	[57]
*Eurhynchium striatum*, *Brachythecium rutabulum*, *Brachythecium salebrosum*, *Scleropodium purum*, *Rhytidiadelphus squarrosus* and *Rhytidiadelphus triquetrus*	FAs (from NL, GL, PL and TL)	HPTLC for NL, PL and GL separation, GC-FID for FA analysis	Acylation (acetyl chloride)	[58]
*Rhytidiadelphus squurrosus* and *Eurhynchium striatum*	FAs	GC-FID	Trimethylsilylation (Triphenylmethanol)	[59]
55 species of *Bryophyta*	FAs on MGDG	TLC for separation, GC for FAs analysis	^-^	[60]
*Marchantia polymorpha*, *Atrichum angustatum* and *Brachythecium* sp.	FAs (from TL, NL, PL, GL)	TLC for separation, GC-MS for FAs analysis	Silylation	[61]
*Calliergon cordifolium*, *Drepanocladus lycopodioides*, *Fontinalis antipyretica* and *Riccia fluitans*	FA (fromTL, TAG, PL, GL)	TLC for separation, GC-MS for FAs analysis	Silylation	[62]
*Marchantia polymorpha*	FAs (from NL, GL, PL)	TLC for separation, HPLC with fluorescence detection	No derivatization	[63]
*Marchantia polymorpha*	FAs	GC-FID	alkylation (BF_3_-methanol	[64]
*Marchantia polymorpha*	FAs	GC-MS	10% methanolic HCl)	[40]
*Marchantia polymorpha*	FAs	GC	Alkylation	[65]
*Physcomitrella patens*	FAs	GC-FID	Alkylation (methylation)	[66]
*Homomallium connexum*, *Actinothuidium hookeri* and *Neckera pennata*	FAs (only carbon number)	HPLC with fluorescence detection, APCI/MS for determination	Alkylation (N,N-dimethyl-formamide)	[67]
*Rhodobryum ontariense*	FAs	GC-MS, NMR	Alkylation (methylation)	[68]
*Atrichum undulatum* and *Hypnum andoi*	FAs	GC-FID and GC-MS	Alkylation (methylation)	[69]
*Physcomitrella patens*, *Encalypta streptocarpa*, *Pottia lanceolata*, *Plagiomnium undulatum*, *Atrichum undulatum*, *Brachythecium rutabulum* and *Rhynchostegium murale*	FAs	GC-MS	Alkylation (methylation)	[13]
*Polytrichum commune* and *Dicranum polysetum*	FAs and other hydrophobic compounds	SPE for separation, GC-MS for identification	Silylation (N,O-Bis(trimethylsilyl)-trifluoroacetamide)	[43]
*Anisothecium spirale*	Free FAs, TAG	HP-TLC for lipids separation, GC-FID and GC-MS for quantification	Alkylation (methylation)	[70]

Abbreviations: HPLC, High-performance liquid chromatography; APCI, Atmospheric pressure chemical ionization; NMR, Nuclear magnetic resonance.

**Table 2 plants-08-00524-t002:** Fatty acids composition in bryophytes.

**Saturated Fatty Acids (SFAs)**																																
**Species**	**Position**	**12:0**	**14:0**	**15:0**		**16:0**	**17:0**			**18:0**			**20:0**				**21:0**				**22:0**			**23:0**		**24:0**			**25:0**		**26:0**	**References**
*Marchantia polymorpha*						24.8 *																				1.2 *						[40]
*Fontinalis antipyretica*	TL	1.5	0.4	1.2		13.6	1.3			5.6			1.0																			[62]
	TAG	0.3	0.7	0.2		4.7				1.8			0.4																			
*Riccia fluitans*	TL	0.2	0.1	0.3		8.5	0.3			1.6			0.2																			
	TAG	0.5	0.2			9.3				3.0			0.3																			
*Pellia neesiana*	TL	0.5	0.3	0.3		12.0	0.5			4.1			0.4																			
	TAG	0.6	0.8	0.5		6.9	0.3			5.0			1.3																			
*Calliergon cordifoliu*	TL	0.7	0.1	0.6		10.2	0.6			3.4			0.4																			
	TAG	1.3	0.1	0.4		12.7	0.3			4.7			2.0																			
*Drepanocladus lycopodioides*	TL	0.8	0.5	0.5		11.0	0.7			6.1			0.6																			
	TAG	0.9	0.7	0.6		18.3	0.4			7.2			1.1																			
*Anisothecium spirale*	PL	2.51 *	16.43 *	2.36 *		44.49 *	1.51 *			14.19 *			1.38 *								1.37 *											[70]
	NL	0.14 *	0.56 *	0.45 *		6.70 *	0.49 *			0.11 *			1.29 *								0.35 *											
*Phycomitrella patens*			0.47	0.47		25.67	0.30			4.24			1.43				0.28				1.36			0.32		0.98			0.12		0.10	[13]
*Encalypta vulgaris*			0.60	0.55		29.97	0.25			1.27			0.78				0.13				0.67			0.34		0.49			0.05		0.04	
*Pottia lanceolata*			0.52	0.43		30.40	0.53			2.05			1.38				0.20				0.93			0.51		2.04			0.26		0.25	
*Plagiomnium undulatum*			0.26	0.24		29.79	0.60			2.70			1.62				0.49				1.52			0.41		0.73			0.04		0.10	
*Brachythecium rutabulum*			0.39	0.33		30.68	0.49			1.11											0.16			0.13		1.15			0.16		0.24	
*Rhynchostegium murale*			0.21	0.16		23.94	0.36			1.13											0.10			0.08		0.99			0.19		0.12	
*Atrichum undulatum*			0.62	0.27		24.80	0.83			1.60			1.66				0.08				0.79			0.20		1.34			0.12		0.09	
*Atrichum undulatum*						22.17				3.34											1.01											[78]
*Hypnum andoi*						63.48				8.08			4.64								6.26					5.16						
*Mnium hornum*						25.25				2.74																						
*Rhodobryum ontariense*						14.31				1.14																						
*Rhytididelphus squarrosus*						11.90				1.00																						[79]
*Sphagnum fimbriatum*			0.90			12.80				0.80																						[80]
*Sphagnum magellanicum*			1.10			14.70				1.60																						
*Sphagnum najus*			1.10			10.70				0.90																						
*Sphagnum nemoreum*			0.70			11.40				2.80																						
**Monounsaturated Fatty Acids (MUFAs)**																																
**Species**	**Position**	**13:1**	**15:1**	**16:1 ^Δ^**	**16:1, n-9**		**16:1, n-7**				**17:1, n-9**				**18:1 ^Δ^**					**18:1, n-11**			**18:1, n-9**			**18:1, n-7**		**C19:1**			**C22:1**	**References**
*Fontinalis antipyretica*	TL	1.5	0.8		2.5		1.7				0.6									1.3			13.4			0.4						[62]
	TAG				1.1		0.6													0.3			10.2			2.2						
*Riccia fluitans*	TL	0.7	0.7		1.1		1.5				0.7									0.9			3.2			0.9						
	TAG				0.4		0.3													0.1			0.5			0.2						
*Pellia neesiana*	TL	0.5	0.4		3.8		2.0				0.6									0.1			10.0			0.9						
	TAG		0.1		2.3		0.6				0.1									0.4			8.4			0.3						
*Calliergon cordifoliu*	TL	0.7	0.6		3.2		1.8				0.6									0.8			6.1			0.8						
	TAG	0.1	0.2		2.6		1.1				0.2									1.3			10.3			1.0						
*Drepanocladus lycopodioides*	TL	0.5	0.4		5.9		2.5				0.6									1.4			10.7			1.0						
	TAG	0.1	0.1		3.9		1.2				0.1									3.2			14.3			0.2						
*Anisothecium spirale*	PL						6.03 *																									[70]
	NL						0.65 *																0.63 *									
*Phycomitrella patens*				0.32											0.48													0.06			0.06	[13]
*Encalypta vulgaris*				0.60											3.61																0.05	
*Pottia lanceolata*				0.43											1.11																0.08	
*Plagiomnium undulatum*				0.38											3.03													0.10				
*Brachythecium rutabulum*				0.17											4.04																0.16	
*Rhynchostegium murale*				0.12											4.85																0.06	
*Atrichum undulatum*				0.46											1.06													1.84			0.08	
*Atrichum undulatum*																							18.49									[78]
*Hypnum andoi*																															12.38	
*Mnium hornum*																							5.13									
*Rhodobryum ontariense*																							2.47									
*Rhytididelphus squarrosus*																							2.30									[79]
*Sphagnum fimbriatum*				0.70											5.70																	[80]
*Sphagnum magellanicum*				0.70											5.90																	
*Sphagnum najus*				2.30											7.70																	
*Sphagnum nemoreum*				1.80											7.80																	
**Polyunsaturated Fatty Acids (PUFAs)**																																
**Species**	**Position**	**16:2**	**16:3 ** **^Δ^**	**16:3, n-6**	**16:3, n-3**		**16:4, n-3**		**18:2 ^Δ^**			**18:2, n-6 (LA)**				**18:2, n-3**			**18:3, n-6 (GLA)**						**18:3 ^Δ^**		**18:3, n-3 (ALA)**				**18:4, n-3**	**References**
*Marchantia polymorpha*					18.4 *							6.4 *							0.7 *								38.9 *				0.5 *	[40]
*Fontinalis antipyretica*	TL											7.7							3.8								9.0					[62]
	TAG											1.3							5.0								6.1					
*Riccia fluitans*	TL			0.6	1.7		2.3					9.1				1.2			5.6								6.2				3.1	
	TAG			0.1	0.3							0.6				0.2			1.2								2.2					
*Pellia neesiana*	TL			0.9	0.9		1.4					6.9				0.9			4.4								9.3				2.6	
	TAG			0.1	0.2							3.1				2.3			1.3								3.9					
*Calliergon cordifoliu*	TL			0.6	1.0		1.1					8.5				0.8			10.9								14.7				3.2	
	TAG			0.2	0.6		1.4					9.8				0.1			11.2								19.7				5.0	
*Drepanocladus lycopodioides*	TL			0.5	0.5		1.0					9.0				0.9			2.9								11.0				2.2	
	TAG			0.3	0.2		0.2					5.6							1.4								16.0				0.6	
*Anisothecium spirale*	PL											0.08 *															9.63 *					[70]
	NL											5.47 *							1 *													
*Phycomitrella patens*		4.29	2.27						24.25																13.27							[13]
*Encalypta vulgaris*		3.66	0.34						32.02																7.66							
*Pottia lanceolata*		3.76	0.17						42.88																4.68							
*Plagiomnium undulatum*		1.34	0.46						24.25																11.32							
*Brachythecium rutabulum*		0.12	0.13						20.51																5.68							
*Rhynchostegiummurale*		0.22	0.00						27.51																8.42							
*Atrichum undulatum*		4.61	0.76						31.49																15.31							
*Atrichum undulatum*												26.80															20.50					[78]
*Mnium hornum*												11.76															19.65					
*Rhodobryum ontariense*												5.25															20.32					
*Rhytididelphus squarrosus*												15.10							1.40								19.10					[79]
*Sphagnum fimbriatum*												22.10															38.20					[80]
*Sphagnum magellanicum*												22.00															34.00					
*Sphagnum najus*												25.90															33.70					
*Sphagnum nemoreum*												30.30															29.20					
**Acetylenic Acids (AFAs)**																																
**Species**	**Position**	**18:1, 6a**	**18:1, 9a**	**18:1, 12a**	**18:2, 6a, 9**			**18:2, 9, 12a**						**18:2, 9a, 12**				**18:3, 6a, 9, 12**				**18:4, 6a, 9, 12, 15**				**20:3, 8a, 11, 14**				**20:4, 5a, 8, 11, 14**		**References**
*Fontinalis antipyretica*	TL	0.6	2.5	1.5	3.2			2.5						0.1				0.1				3.1				0.1				0.5		[81]
	TAG	2.0	9.4	4.4	16.3			9.0						1.9				2.2				15.2				1.4				1.2		
*Riccia fluitans*	TL	0.8	1.6	1.5	2.9			1.1						0.9				5.3				1.0				1.4				2.6		
	TAG	1.4	7.0	6.7	9.9			4.7						3.4				24.2				5.1				6.3				11.5		
*Pellia neesiana*	TL	0.1	0.6	0.5	0.4			0.4						0.4				3.1				1.4				0.5				0.4		
	TAG	2.7	5.8	5.4	2.6			4.1						4.1				14.1				9.2				5.2				3.8		
*Calliergon cordifoliu*	TL	0.1	0.1	0.1	0.1			0.1						0.1				0.1				0.3				0.1				0.1		
	TAG	0.1	0.1	0.2	0.3			0.6						0.8				0.3				3.3				0.5				0.4		
*Drepanocladus lycopodioides*	TL	0.1	0.2	0.3	0.1			0.2						0.2				1.2				0.6				0.1				0.2		
	TAG	1.1	1.2	2.1	0.8			1.2						1.1				6.3				3.1				1.0				1.1		
*Anisothecium spirale*	PL																					0.5 *										[70]
	NL																	1.9 *				72.19 *										
*Rhodobryum ontariense*																		13.3				42.3										[68]

Abbreviations: PL, Polar lipid; NL, Neutral lipid; TL, Total lipid; TAG, Triacylglycerol; Unit: % of total FAs. * mol% of total FA. **^Δ^** fatty acids with no specific positional information. Species in blue are liverworts whereas mosses are in green. The fatty acid composition and content are very much depending on the growth condition, this table summarizes information from various conditions, and therefore it is only considered as a guidance.

**Table 3 plants-08-00524-t003:** Bryophytes species with high contents of AA and EPA.

Species	Tissues	Regions	AA Contents *	EPA Contents *	References
*Eurhynchium striatum*	Gametophyte	Switzerland	36.7	10.8	[58]
*Brachythecium rutabulum*			23.5	23.4	
*Brachythecium salebrosum*			20.9	15.0	
*Scleropodium purum*			29.0	8.6	
*Rhytidiadelphus squarrosus*			24.0	14.9	
*Rhytidiadelphus triquetrus*			24.6	9.5	
*Eurhynchium striatum*	Protonema	Collected in Switzerland then keep in cell culture	33.6	2.7	
*Brachythecium rutabulum*			40.1	4.7	
*Brachythecium salebrosum*			32.2	4.9	
*Scleropodium purum*			41.5	2.6	
*Rhytidiadelphus squarrosus*			32.4	5.2	
*Rhytidiadelphus triquetrus*			20.0	1.7	
*Marchantia polymorpha*		Cell culture	11	3	[63]
*Marchantia polymorpha*		Cell culture	2.2 ^+^	2.6 ^+^	[14]
*Marchantia polymorhpa*		Agar plate, WT	3.1 ^#^	5.9 ^#^	[40]
*Leptobryum pyriforme*	Protonema	Cell culture	20	7	[84]
*Physcomitrella patens*	Gametophores	Cell culture	18.7	1.5	[13]
	Protonema		15.9	6.8	
*Rhynchostegium murale*	Gametophores	Cell culture	26.4	3.5	
*Mnium cuspidatum*	Gametophores	Minnesota	11.4	8.9	[85]
*Mnium medium*	Gametophores	Minnesota	23.0	19.0	
*Hylocomium splendens*	Gametophores	Alaska	12.9	18.3	
*Pleurozium schreberi*	Gametophores	Alaska	29.0	11.0	
*Rhytididelphus squarrosus*	Gametophores	Germany	30.7	14.4	[86]
*Atrichum undulatum*		Germany	6.21	1.52	[69]
*Mnium hornum*			26.03	9.44	[78]
*Anisothecium spirale*	Gametophyte	Eastern Himalayas	1.09 ^#^	0.27 ^#^	[70]

* % of total FAs; ^#^ mol% of total FAs; ^+^ mg/g dry cells.

**Table 4 plants-08-00524-t004:** Environmental factors affecting AA and EPA content in bryophytes, as well as genetic modification that result in high AA and EPA content.

**Environmental Stresses**							
**Species**	**Growtd Conditions**	**Environmental Factors**	**Variables**	**Biomass Growtd (mg/plant)**	**AA ***	**EPA ***	**References**
*Rhytidiadelphus squarrosus*	MS medium	Temperature	5 °C		18.6	9.4	[59]
			10 °C		23.8	13.7	
			15 °C		24.6	11.7	
			20 °C		32.7	5.7	
			25 °C		26.8	6.3	
			30 °C		23.0	2.4	
*Eurhynchium striatum*	MS medium	Temperature	5 °C		28.4	5.6	
			10 °C		32.5	4.9	
			15 °C		30.5	7.5	
			20 °C		34.1	2.9	
			25 °C		32.9	3.0	
			30 °C		31.7	2.4	
*Rhytidiadelphus squarrosus*	MS medium	pH	5.8		13.2		
			6.5		9.5		
*Marchantia polymorpha*	MS medium	Temperature	15 °C		12.1	3.6	[63]
			25 °C		11	3	
*Marchantia polymorpha*	M51C solid medium	Temperature	5 °C		2	14	[88]
			25 °C		3.5	5	
*Ctenidium molluscum*		Nitrogen (g/L)	0		22.9	5.3	[89]
			0.04		14.7	2.6	
			0.4		17.5	5.0	
*Pogonatum urnigerum*			0		2.8	0.4	
			0.04		3.0	3.9	
			0.4		1.7	1.1	
*Dichodontium pellucidum*			0		1.7	1.1	
			0.04		3.4	2.2	
			0.4		3.3	1.9	
*Tortella tortuosa*			0		5.1	1.8	
			0.04		5.3	1.5	
			0.4		5.8	1.9	
*Marchantia polymorpha*		Photon flux density (umol/m^2^/s)	3	1028.2 ^+^	2.8 ^+^	2.6 ^+^	[64]
			9	979.0 ^+^	2.9 ^+^	2.8 ^+^	
			20	976.8 ^+^	3.7 ^+^	3.4 ^+^	
			32	987.0 ^+^	3.3 ^+^	3.1 ^+^	
		Osmolarity (NaCl %)	0	576.5 ^+^	3.0 ^+^ (C_20_ PUFA)		
			0.2	651.6 ^+^	3.0 ^+^ (C_20_ PUFA)		
			0.5	62.9 ^+^	0.2 ^+^ (C_20_ PUFA)		
*Marchantia polymorpha*		Light quality	White	543	Not reported		[65]
			Blue	454	Varied slightly	1.5-fold higher than under white light	
		Light intensity (umol/m^2^/s)	40	696	5.7	2.5	
			60	1075	5.7	3.0	
			80	1088	5.7	4.5	
**Genetic Transformation**							
**Species**	**Types**		**AA ***		**EPA ***		**Reference**
*Marchantia polymorpha*	WT		3.1		5.9		[40]
	DEOE-34		5.0		12.1		
	DEDOE-58		11.4		8.9		

MS: Murashige—Skoog medium; WT, wild type; DEOE-34, overexpression of Δ-6 desaturase and Δ-6 elongase; DEDOE-58, overexpression of Δ-5 desaturase, Δ-6 desaurase and Δ-6 elongase. * % of total lipids; ^+^ mg/L/Day.

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
