# Peer review of "Valuable Fatty Acids in Bryophytes—Production, Biosynthesis, Analysis and Applications"

_plants, 2019, doi:10.3390/plants8110524_

Round 1
Reviewer 1 Report
The review by Lu et al. gives a thorough overview of the lipids in bryophytes. The authors summarized the fatty acid compositions in bryophytes, their biosynthetis pathways, various strategies of enhancing VL-PUFAs production, identification and quantification methods. However, there are concerns that should be addressed to strengthen the manuscript.
Major concerns:
“Methods of lipids and fatty acids analysis” is relatively weak for a review article.
Line 235: “solvents with different polarities are used for separating different lipid classes” is very vague since the examples of “solvents with different polarities” were not mentioned anywhere in this review.
Line 239 : “large amount of sample”. Please be more specific. If the amount of samples used in lipid analysis is an issue, it should be discussed here (e.g., how many milligrams of tissues are normally used for lipid analysis). Perhaps a table summarizing LOD and LOQ of each method could be included here.
Please discuss internal standards used in different methods of quantification.
When mass spectrometry is used for quantification, please specify if total ion chromatogram (TIC) or selected ion monitoring (SIM) was used. When SIM is used, characteristic ions should be discussed here as well.
If the FAs were derivatized, please describe how the derivatizations were performed e.g. silylation, acylation, acetylation. If the methods did not require derivatization of FAs, please state that also.
Minor concerns:
Table 4: Please replace “mass spectroscopy” with “mass spectrometry”.
polyunsaturated fatty acids, arachidonic acid and eicosapentaenoic acid should be written vl-PUFAs, AA and EPA, respectively after they first mentioned.
Author Response
Dear Reviewer:
Our responses to the reviewers are given in red, below the relevant text. We are very grateful for the great suggestions by you and feel that the text after restructuring the manuscript the text is much easier to read.
Reviewer 1:
The review by Lu et al. gives a thorough overview of the lipids in bryophytes. The authors summarized the fatty acid compositions in bryophytes, their biosynthetis pathways, various strategies of enhancing VL-PUFAs production, identification and quantification methods. However, there are concerns that should be addressed to strengthen the manuscript.
Major concerns:
“Methods of lipids and fatty acids analysis” is relatively weak for a review article.The title has been changed to “Analysis of fatty acids in bryophytes.
Line 235: “solvents with different polarities are used for separating different lipid classes” is very vague since the examples of “solvents with different polarities” were not mentioned anywhere in this review.We have added additional information to these parts.
Line 239: “large amount of sample”. Please be more specific. If the amount of samples used in lipid analysis is an issue, it should be discussed here (e.g., how many milligrams of tissues are normally used for lipid analysis). Perhaps a table summarizing LOD and LOQ of each method could be included here.The text on the lipid analysis is updated, to also include comments on problems with obtaining pure samples. We do not find that additional reviews on LOD and LOQ are needed in this manuscript, but we have referred the reader to many other reviews.
Please discuss internal standards used in different methods of quantification.This is a difficult topic. Most of the papers mentioned in this review were published before 2000, and in most of the papers there were either no internal standards used or the description was vague.
When mass spectrometry is used for quantification, please specify if total ion chromatogram (TIC) or selected ion monitoring (SIM) was used. When SIM is used, characteristic ions should be discussed here as well.For the same reason as mentioned above. Likewise, for most of the papers there was no information about the ion mode, thus we only added the method of derivatization in the table as other information would be very sketchy.
If the FAs were derivatized, please describe how the derivatizations were performed e.g. silylation, acylation, acetylation. If the methods did not require derivatization of FAs, please state that also.The derivatization methods have been added.
Minor concerns:
Table 4: Please replace “mass spectroscopy” with “mass spectrometry”.
polyunsaturated fatty acids, arachidonic acid and eicosapentaenoic acid should be written vl-PUFAs, AA and EPA, respectively after they first mentioned.
All the minor concerns have been addressed.
Reviewer 2 Report
I find this manuscript very solid. I recommend moderate English edition, but I have no other comments.
Author Response
Dear reviewer,
The English has been updated throughout the manuscript. We are very grateful for your great suggestions and feel that the updated manuscript is much easier to read.
Reviewer 3 Report
Dear Sir or Madam,
the manuscript „Valuable fatty acids in bryophytes – production, biosynthesis, analysis and applications“ represents a review work, addressing the fatty acid metabolome of bryophytes. The authors did a good job, and collected a huge body of information, quite useful for the people, working in the field of fatty acid metabolomics. From this point, of course, such a review would be quite interesting for publishing in plants. However, there are some issues with this work, which need to be solved before the manuscript is published.
Thus, to be published, the following major concerns need to be addressed:
The quality of English needs to be improved. The current state, to my mind, is inacceptable for publication. The manuscript needs to be re-structured, the “red line” of the manuscript needs to be built and clearly seen. I would suggest. To start with the principle scheme of FA metabolism in plants, then to show differences observed in Bryophytes, then to discuss these differences in the context of the data on algae metabolism. After this I would put methods, and then go to specific groups of FAs one by one. When describing methods of FA analysis, address also some new ones, also those, which still were not used in bryophyte metabolomics. Probably, these methods are still unknown for these people, and you could increase the novelty of your work. For example, MALDI-TOF-MS was not mentioned by the authors: Podolskaya et al. Thin film chemical deposition techniques as a tool for fingerprinting of free fatty acids by MALDI-TOF-MS. Anal Chem. 91(2):1636-1643. Please, take care, that all abbreviations are explained by the first use (you give the full meaning and then the abbreviation itself in brackets), and make please a comprehensive abbreviation list.I think, all other points need to be addressed in the second round after the revision of the issues listed above, as these might be mostly related to the English quality
Author Response
Dear Reviewer,
Our responses are given in red below the relevant text. We are very grateful for the great suggestions by you and feel that the text after restructuring the manuscript the text is much easier to read.
Reviewer 3:
Dear Sir or Madam,
The manuscript „Valuable fatty acids in bryophytes – production, biosynthesis, analysis and applications“ represents a review work, addressing the fatty acid metabolome of bryophytes. The authors did a good job, and collected a huge body of information, quite useful for the people, working in the field of fatty acid metabolomics. From this point, of course, such a review would be quite interesting for publishing in plants. However, there are some issues with this work, which need to be solved before the manuscript is published.
Thus, to be published, the following major concerns need to be addressed:
The quality of English needs to be improved. The current state, to my mind, is inacceptable for publication. The manuscript needs to be re-structured, the “red line” of the manuscript needs to be built and clearly seen. I would suggest.
To start with the principle scheme of FA metabolism in plants, then to show differences observed in Bryophytes, then to discuss these differences in the context of the data on algae metabolism. After this I would put methods, and then go to specific groups of FAs one by one.
The structure of the manuscript has been changed. The biosynthesis chapter is now the first one after introduction, followed by methods for analysis, then FAs found in bryophytes. We combined the chapter acetylenic acid into FAs found in bryophytes. Following this the English has been corrected throughout the manuscript.
When describing methods of FA analysis, address also some new ones, also those, which still were not used in bryophyte metabolomics.
Probably, these methods are still unknown for these people, and you could increase the novelty of your work. For example, MALDI-TOF-MS was not mentioned by the authors: Podolskaya et al. Thin film chemical deposition techniques as a tool for fingerprinting of free fatty acids by MALDI-TOF-MS. Anal Chem. 91(2):1636-1643.
We have added a paragraph on this method. As most people that study FAs in bryophytes come from chemical fields within analytical chemistry, we do not have to tell them about the latest methods. The main reason for the low number of studies is simply due to the difficulties of getting pure samples.
Please, take care, that all abbreviations are explained by the first use (you give the full meaning and then the abbreviation itself in brackets), and make please a comprehensive abbreviation list.
We did not find instruction about where to include the list of abbreviation, however, a list of abbreviations was added to the end of the document.
I think, all other points need to be addressed in the second round after the revision of the issues listed above, as these might be mostly related to the English quality.
The English has been updated throughout the manuscript.
Round 2
Reviewer 1 Report
The revised version of the review by Lu et al. showed significant improvements and addressed most of the concerns raised. Minor spell checks will be beneficial throughout the manuscripts. The overall quality warrants publication in Plants.